# Impact of *Azospirillum* sp. B510 on the Rhizosphere Microbiome of Rice under Field Conditions

**Michiko Yasuda [1,\*], Khondoker M. G. Dastogeer [1,2], Elsie Sarkodee-Addo [1], Chihiro Tokiwa [1], Tsuyoshi Isawa [3], Satoshi Shinozaki [3] and Shin Okazaki [1]**

[1] Plant Microbiology Laboratory, Tokyo University of Agriculture and Technology, Saiwaicho 3-5-8, Fuchu-shi, Tokyo 183-8509, Japan; dastogeer.ppath@bau.edu.bd (K.M.G.D.); elsieaddo67@yahoo.com (E.S.-A.); s218271t@st.go.tuat.ac.jp (C.T.); sokazaki@cc.tuat.ac.jp (S.O.)

[2] Department of Plant Pathology, Bangladesh Agricultural University, Mymensingh 2202, Bangladesh

[3] Mayekawa Research Institute Co., Ltd. Botan 3-14-15, Koto-ku, Tokyo 135-8482, Japan; tsuyoshi-isawa@mayekawa.co.jp (T.I.); satoshi-shinozaki@mayekawa.co.jp (S.S.)

[\*] Correspondence: ysdmichi@cc.tuat.ac.jp; Tel.: +81-42-367-5847

**Abstract:** There has been increasing attention toward the influence of biofertilizers on the composition of microbial communities associated with crop plants. We investigated the impact of *Azospirillum* sp. B510, a bacterial strain with nitrogen-fixing ability, on the structure of bacterial and fungal communities within rice plant rhizospheres by amplicon sequencing at two sampling stages (the vegetative and harvest stages of rice). Principal coordinate analysis (PCoA) demonstrated a significant community shift in the bacterial microbiome when the plants were inoculated with B510 at the vegetative stage, which was very similar to the effect of chemical N-fertilizer application. This result suggested that the inoculation with B510 strongly influenced nitrogen uptake by the host plants under low nitrogen conditions. Least discriminant analysis (LDA) showed that the B510 inoculation significantly increased the $N_2$-fixing *Clostridium*, *Aeromonas* and *Bacillus* populations. In contrast, there was no apparent influence of B510 on the fungal community structure. The putative functional properties of bacteria were identified through PICRUSt2, and this hinted that amino acid, sugar and vitamin production might be related to B510 inoculation. Our results indicate that B510 inoculation influenced the bacterial community structure by recruiting other $N_2$-fixing bacteria in the absence of nitrogen fertilizer.

**Keywords:** *Azospirillum* sp. B510; paddy field; rhizosphere; nitrogen fertilizer; microbiome; cobalamin





## 1. Introduction

Biofertilizers have been widely used in many countries as alternatives to chemical fertilizers to increase soil fertility and crop production for sustainable farming. The application of beneficial microbes such as *Rhizobium*, *Azobacter*, *Azospirillum*, *Bacillus*, *Pseudomonas* and *Mycorrhiza* can enhance plant growth and the resistance to adverse environmental stresses, such as water and nutrient deficiencies and heavy metal contamination [1]. Plant-growth-promoting rhizobacteria (PGPR) are among these microbes, some of which are commercially used as biofertilizers [2]. High-throughput sequencing technologies have provided new findings about how microbial communities shift after biofertilizer application [3–7].

*Azospirillum* is a genus of alpha-proteobacteria in the family *Azospirillaceae*, a well-studied member of PGPR found in association with some of the world's most staple food crops, including rice, maize, sorghum, wheat and millet [8–10]. Members of the genus *Azospirillum* are widespread in soil, and their inoculation on cereals and forage crops results in yield increases in many field experiments, not only due to nitrogen fixation but also through the production of plant-growth-promoting substances, such as the phytohormones indole-3 acetic acid (IAA) and gibberellic acid [11,12]. The inoculation of *A. brasilense* onto *Zea mays* and *Sorghum bicolor* has been reported to enhance the uptake of mineral ions [13].

$NH_4^+$ and $PO_4^-$ uptake was also enhanced in rice plants after the inoculation with *A. lipoferum* under hydroponic conditions [14].

*Azospirillum* sp. B510 was isolated in Japan from rice stems and can colonize the surface and interior tissues of rice roots [15,16]. Increased plant growth measured by tiller number and seed production after B510 colonization was demonstrated under greenhouse, paddy field and laboratory conditions [16–19]. Kaneko et al. [20] determined the complete genome sequence of B510; the organism has a single chromosome and six plasmids encoding 3416 putative proteins, including $N_2$-fixation-related genes. B510 induces a modified metabolic and transcriptomic response in shoots and roots, suggesting that this bacterium triggers a systemic response against pathogens [16,21–23]. However, the combination of the rice cultivar and microbial strain is important for the PGPR activity [16,21]. Only one paper has shown a slight change in the bacterial community structure between B510-inoculated rice plants and non-inoculated rice plants [3]. Neither the interaction between B510 and other microorganisms in the rhizosphere nor the relationship among B510 and B510-related microorganisms and host roots has been elucidated.

To clarify the effects of inoculation with *Azospirillum* sp. B510 and nitrogen fertilizers on plant growth and the diversity of the bacteria and fungi associated with rice roots, we conducted a field experiment. Next-generation sequencing was used to evaluate the bacterial and fungal community diversity in the rhizosphere of rice plants under different treatments. We also compared two sampling stages (the vegetative and harvest stages of rice). The Linear Discriminant Analysis (LDA) Effect Size (LEfSe) analysis demonstrated that specific bacteria and fungi were recruited to the rice rhizosphere of B510-inoculated plants, including some $N_2$-fixing bacteria. Furthermore, predicted function analysis using PICRUSt2 showed that the recruited bacteria might contribute to the supply of vitamins that plants cannot produce. Our work expanded the knowledge of B510's ability as a biofertilizer to influence microbial interactions in the rhizosphere, providing fundamental knowledge about the rhizosphere in ecofriendly agriculture.

## 2. Materials and Methods

### 2.1. Field Experiments

The field experiments in this study were conducted in an experimental field at the Fuchu-honmachi paddy field belonging to the Tokyo University of Agriculture and Technology (35°41′ N, 139°29′ E, 46 AMSL) in Japan. Rice seeds (*Oryza sativa* cv. Nipponbare) were sterilized using 0.1% (*v/v*) Sumithion and 0.5% (*v/v*) Sportac Stana SE (Sumitomo Chemical, Tokyo, Japan) for 24 h. After washing with tap water five times consecutively, the seeds were soaked in tap water for germination. The seeds were sown in a granular culture soil (Kumiai-Ryuujou-Baido; Sun Agri Co., Ltd., Tokyo, Japan) on 20 May 2019.

For inoculations with *Azospirillum* sp. B510, we used the commercial bacterial solution Ine-Fighter® (Mayekawa Co. Ltd. Tokyo, Japan); the commercial bacterial solution (3.3 mL) was diluted in 1 L sterilized distilled water. Young rice seedlings grown in nursery boxes (approximately $2 \times 10^7$ CFU/mL) were sprayed with 1 L of the diluted bacterial solution two days before transplantation, following commercial instructions.

At the fourth leaf stage, the seedlings were transplanted to the paddy field on 5 June 2019. The planting density was 120 m$^{-2}$, with a spacing of 30 cm $\times$ 30 cm. The field was submerged throughout the experiments by irrigation water. After transplantation, $P_2O_5$ and $K_2O$ were applied at 30 kg ha$^{-1}$ as basal fertilizers. A total of 20 kg ha$^{-1}$ of $NH_4SO_4$ was applied in five treatments at two-week intervals.

### 2.2. Evaluation of Plant Growth and Yield

To evaluate the effect of B510 on the rice plant growth, we measured the plant height, the plant weight, the number of tillers and panicles at the vegetative stage 63 days after transplantation (DAT) and the grain weight at the harvest stage (112 DAT).

### 2.3. Physicochemical Properties of Soil

The measurements of the soil chemical properties were carried out at the physico-chemical analysis center of Vegetech Co. Ltd. (Kanagawa, Japan). The soil pH and electric conductivity (EC) were determined in deionized water ($H_2O$) and 1 M KCl at a soil-to-solution ratio of 1:5. Total carbon (TC) and total nitrogen (TN) were quantified using the dry combustion method [24].

Subsequently, the suspension was centrifuged and filtered through filter paper (Advantec No. 5C; Advantec, Tokyo, Japan). Several drops of 1 mg $kg^{-1}$ of a copper (II) bromide solution were added to the extract and kept at 4 °C. The N-$NH_4$ in the extract was analyzed by the modified indophenol blue method (Rhine et al., 1998) using a spectrophotometer (UV-1200; Shimadzu Co., Ltd., Kyoto, Japan). N-$NO_3$ in the extract was analyzed by flow injection analysis using a flow-through visible spectrophotometer (S/3250; Soma-Kogaku Co., Ltd., Tokyo, Japan) connected to a double-plunger pump (PE-230; Aqua-lab Co., Ltd., Tokyo, Japan) equipped with a copperized cadmium column to reduce nitrate to nitrite in the sample solutions. The available phosphate (Available-P) was evaluated by the Bray 2 method (Nanzyo, 1997).

### 2.4. DNA Extraction and Amplicon Analysis

DNA extraction was carried out according to a previously published method [25]. The rice roots were collected at 63 and 112 days after transplantation (DAT) on Aug 7 and Sep 25, 2019. Three plant roots were combined as a single sample. Amplicon sequencing of the bacterial V3/V4 region and the fungal internal transcribed spacer (ITS) region was carried out with the MiSeq platform at the Bioengineering Lab. Co. (Sagamihara, Japan). The primers V3/V4f_MIX (ACACTCTTTCCCTACACGACGCTCTTCCGATCT-NNNNN–CCTACGGGNGGCWGCAG) and V3/V4r_MIX (GTGACTGGAGTTCAGACGTGTGCTC-TTCCGATCT-NNNNN-GACTACHVGGGTATCTAATCC) were used to amplify the bacterial V3/V4 region (341f-805r). The first PCR reaction was prepared in a final volume of 20 μL comprised of the DNA template (10 ng) using the ExTaq HS polymerase kit (Takara). The PCR conditions were employed according to the manufacturer's instructions. The PCR products were purified using AMPure XP (Beckman Coulter) and quantified using Synergy H1 (Bio Tek) and the QuantiFluor dsDNA system (Promega). Library quality checks were accomplished using the Fragment Analyzer and dsDNA 915 Reagent Kit (Advanced Analytical Technologies). The libraries were pooled and loaded into the Illumina MiSeq instrument following the manufacturer's instructions (Illumina, San Diego, CA, USA).

The Quantitative Insights into Microbial Ecology (Qime 2.0) toolkit was used to process the raw high-throughput sequencing data. The sequencing data from the 16S and ITS amplicon were analyzed using the R software Ampvis2 package and MicrobiomeAnalyst [26], and the operational taxonomic units (OTUs) were annotated as the EzBioCloud 16S database for bacteria and as UNITE (ver. 8.2) for fungi. The amplicon data were subjected to rarefaction curve, microbial alpha- and beta-diversity and relative abundance analyses using the ranacapa and phyloseq packages. The preferential microbial taxa were compared across treatments using Linear Discriminate Analysis (LDA) Effect Size (LEfSe ver 1.0.8) analysis, with the LDA score >4.0 [27]. The raw sequencing reads were submitted to the DRA/DDBJ under accession no. DRA014017. The bacterial functions were predicted by PICRUSt2 software (ver. 2.3.0 b) based on the KEGG functional database [28].

### 2.5. Statistical Analysis

The statistical analyses were performed using a one-way analysis of variance (ANOVA) followed by the Student–Newman–Keuls (SNK) test and an unpaired two-tailed Student's t-test using SPSS Statistics 28.0 software.

## 3. Results

### 3.1. The Effect of Azospirillum sp. B510 Inoculation on the Growth and Yield of Rice

We tested the plant-growth-promoting activity of B510 in paddy fields where supplemental nitrogen had been applied (+N) and where nitrogen had not been applied (−N). We evaluated the growth of rice at the vegetative stage and harvest stage for plant height, weight, the number of tillers and grain weight (Table 1). Overall, the nitrogen application (+N) treatment resulted in better growth than the condition with supplemental nitrogen (−N) at both the vegetative and harvest stages.

**Table 1.** Effect of *Azospirillum* sp. B510 inoculation on the plant growth.

| Vegetative Stage (63 DAT) | Non-Applied Nitrogen Fertilizer (−N) | | Applied Nitogen Fertilizer (+N) | |
|---|---|---|---|---|
| | Non-Inoculated | Inoculated with B510 | Non-Inoculated | Inoculated with B510 |
| Shoot length (cm/plant) | 86 ± 5.1 ab | 83 ± 3.8 b | 91 ± 3.5 a | 87 ± 3.3 a |
| Shoot weight (g/plant) | 319 ± 47 abd | 297 ± 113 b | 368 ± 50 ab | 387 ± 65 a |
| Root length (g/plant) | 25.3 ± 4.1 a | 22.6 ± 4.1 a | 24.7 ± 2.4 a | 23.2 ± 2.89 a |
| Root weight (g/plant) | 123 ± 29 a | 113 ± 48 a | 121 ± 22 a | 140 ± 20 a |
| **Harvest Stage (112 DAT)** | **Non-Applied Nitrogen Fertilizer (−N)** | | **Applied Nitogen Fertilizer (+N)** | |
| | Non-Inoculated | Inoculated with B510 | Non-Inoculated | Inoculated with B510 |
| Shoot length (cm/plant) | 103 ± 5.5 a | 105 ± 4.4 a | 108 ± 4.4 a | 105 ± 4.5 a |
| Shoot weight (g/plant) | 257 ± 57.4 b | 307 ± 33.2 ab | 348 ± 64.8 ab | 370 ± 63.4 a |
| Root weight (g/plant) | 84 ± 25.5 a | 96 ± 12.7 a | 91 ± 15.6 a | 102 ± 31.6 a |
| Tiller number (number/plant) | 20.7± 2.5 b | 25.7 ± 2.8 a | 27.8 ± 4.4 a | 30.7 ± 7.05 a |
| Panicle number (number/plant) | 19.9 ± 4.5 c | 24.7 ± 3.3 b | 25.9 ± 3.8 ab | 29.9 ± 6.5 a |
| Grain weight (g/plant) | 53.6 ± 14.4 b | 69.3 ± 10.0 ab | 69.2 ± 16.4 a | 73.5 ± 14.2 a |

Effects of B510 on plant growth characteristics (shoot length, shoot weight, root length, root weight, tiller number, panicle number and grain weight) in the paddy field conditions with or without nitrogen fertilizer. These rice characters were measured 63 and 112 days after transplanting. All values are the average ± SD from three sets of three plants each. Different letters indicate statistically significant differences between treatments (Student–Newman–Kuels [SNK] test, $p < 0.05$).

The soil chemical properties before and after rice cultivation showed that the amount of N-NH$_4$ in the +N treatment (0.70 mg/100 g) was slightly higher than that in the −N treatment (0.57 mg/100 g) in the soil after harvest (Table S1). The rice plants inoculated with B510 also tended to have increased biomass and tiller and panicle numbers (Table 1). The number of tillers and the number of panicles in the −N condition were significantly higher when B510 was inoculated at the harvest stage (Table 1). These results indicate that inoculation with B510 had a plant-growth-promoting ability in the non-fertilized condition.

### 3.2. The Effect of B510 Inoculation on Bacterial Community Structure

A total of 2,245,130 bacterial 16S rRNA gene sequences and 4,140,226 fungal ITS gene sequences were generated after filtering, quality control and the removal of chloroplast and mitochondrial OTUs (Table S2). The bacterial and fungal rarefaction curves reached saturation at ~50,000 sequences (Figure S1).

The $\alpha$-diversity of the bacterial community was calculated based on microbial diversity (Shannon index) and richness (Chao1 index) at the OTU level (Figure 1A, Table S3). At

the vegetative stage, the bacterial diversity in B510-inoculated plants was significantly increased compared to that in non-inoculated plants growing in −N conditions, despite no apparent differences in plant growth (Figure 1A, Table 1 and Table S3). For a better understanding of the impact of B510 inoculation on the microbial community, PCoA was used to examine beta-diversity (Figure 1B). At the vegetative stage, the B510-inoculated samples were more similar to the +N samples than the non-inoculated samples (Figure 1B). Similar to Figure 1A, the samples from the harvest stage were very similar and were distinct from the samples of the vegetative stage (Figure 1B). The relative abundances of the top 20 bacterial classes are shown in the heat map (Figure 2), which reveals that proteobacterial abundance was characteristic of the vegetative and harvest stages. The most abundant taxa were beta-proteobacteria in the +N condition at the vegetative stage. The relative abundance of beta-proteobacteria increased from 11.8% to 27% after B510 inoculation in the absence of supplemental nitrogen (−N) conditions during the vegetative stage. Consistently, the second-highest abundance of delta-proteobacteria was reduced by inoculation with B510 at the vegetative stage. In the harvest stage, no significant differences in proteobacterial abundance were observed (Figure 2).

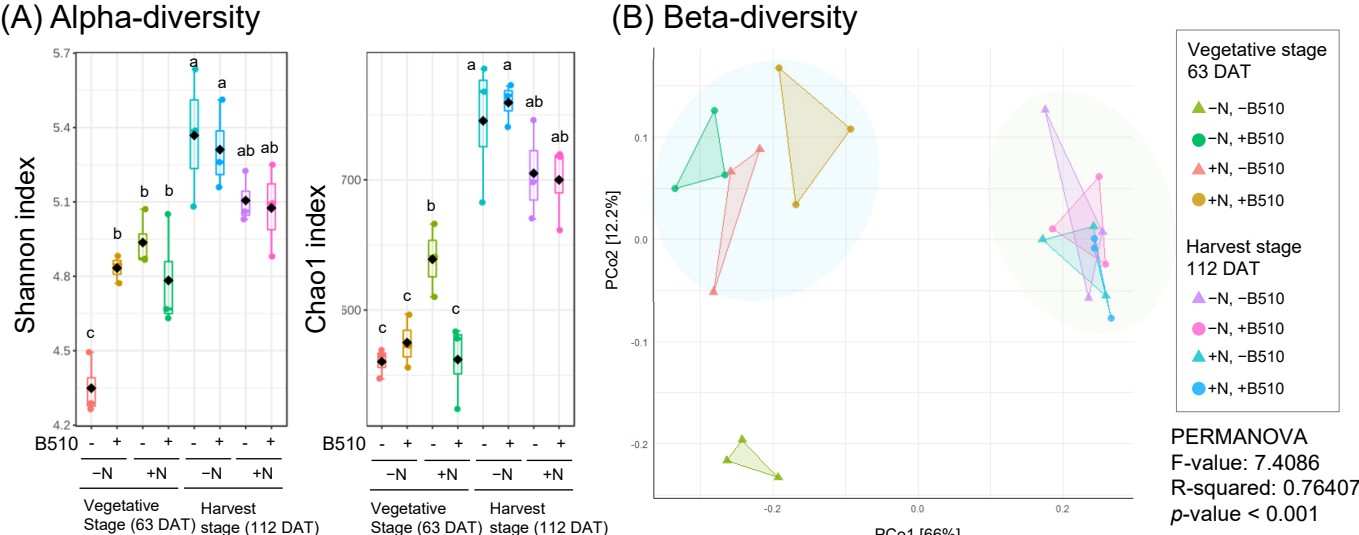

**Figure 1.** Influence of B510 on root-colonized microbiota with different levels of N-fertilizer and two stages of plant development. Measurement of the alpha-diversity (**A**) and beta-diversity (**B**) of the rice root-colonized bacteria. Rice plants inoculated with B510 (+) and without B510 (−), and fertilized with chemical nitrogen (+N) and non-fertilized (−N) harvested at the vegetative stage (63 DAT) and the harvest stage (112 DAT). (**A**) Shannon index and Chao1 index were used in the analysis of the alpha diversity. All values are means of three replicates ± standard errors. Statistical analysis was performed, different letters indicate significant difference between treatments (Student-Newman-Keuls [SNK] test, $p < 0.05$, n = 3). (**B**) Principal Coordinate Analysis (PCoA) are plotted based on Bray-Curtis distance metrices for taxonomical data ($p < 0.01$). Permutational multivariate analysis of variance (PERMANOVA) was performed.

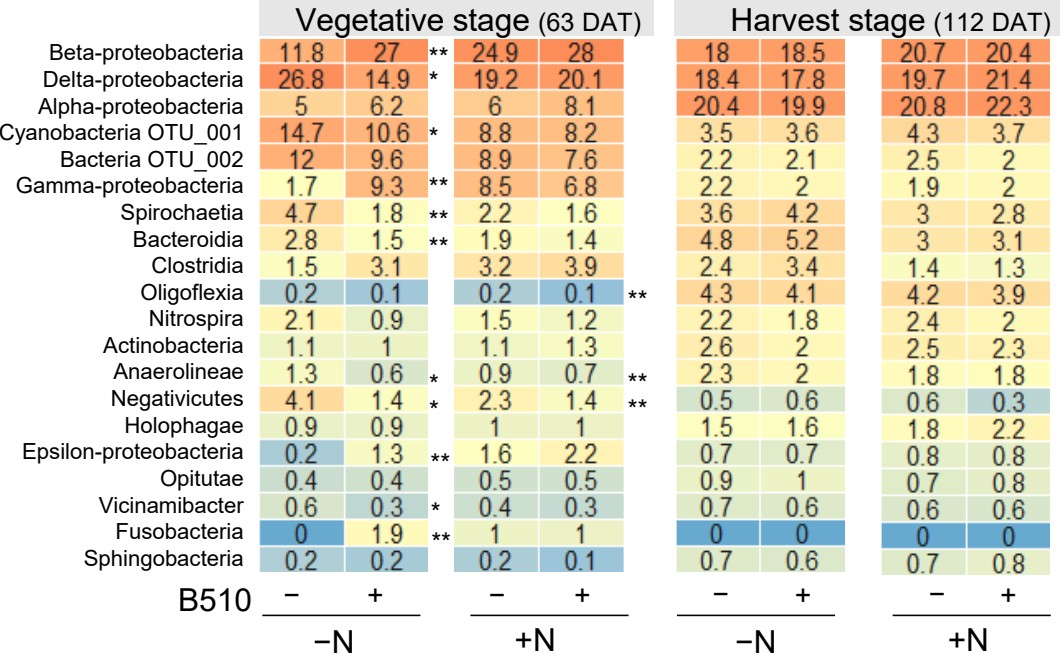

**Figure 2.** Heat Map of the top 20 bacterial species associated with rice roots at the vegetative stage (63 DAT) and harvest stage (112 DAT). The heat-maps show the relative abundances of bacterial operational taxonomic units (OTUs) at the class level from rice plants inoculated with B510 (+) and without B510 (−) and with supplemental nitrogen fertilizer (+N) and without N-fertilizer (−N). The color of the heat-map indicates the relative abundance from high (red) to low (blue). The asterisks identify significant differences between B510-inoculated (+) and without B510 (−) analyzed by the unpaired two-tailed Student's t-test (* $p < 0.05$, ** $p < 0.01$).

We used LEfSe to determine the differences in taxa abundance with or without B510 inoculation. The cladogram depicts the basal relationships among the microbial clades affected by B510 at a high resolution (Figure 3A). The positive and negative bacterial coefficients (LDA > 3) are indicated in Figure 3B. The LEfSe analysis of the bacteria showed that *Clostridium*, *Uliginosibacterium*, *Rhodoferax*, *Exiguobacterium*, *Bacillus*, *Sphingomonas* and *Niveibacterium* significantly increased in the rice rhizosphere inoculated with B510 (Figure 3B). Consistently, *Desulfovibrio* and *Labilithrix* were less abundant in the B510-inoculated rhizosphere (Figure 3B).

### 3.3. The Predicted Function of Bacteria Influenced by B510 Inoculation

The bacterial community structures were significantly different between the non-inoculated plants and B510-inoculated plants in non-nitrogen (−N) conditions (Figure 2B). The PICRUSt2 algorithm provides interoperability with any OTU-picking or denoising algorithm and enables phenotype predictions from a larger database of gene families and reference genomes (Douglas et al., 2020). A total of 424 pathways were selected by PI-CRUSt2 (Figure S2, Table S4). This list identified amino acid production (arginine, lysine, histidine, methionine, isoleucine, leucine, glycine, alanine, glutamate, valine), vitamin production (B1, B6, B9, B12, E), bacterial iron-chelating agent siderophores such as aerobactin and polyamines as the major pathways affected by the treatments (Figure 4A, Table S4). Moreover, the mean real frequency related to nitrogen compound metabolism increased with B510 inoculation (Figure 4B, Table S4). These results indicate that the bacteria colonizing near the rice rhizosphere increased with B510 inoculation, activating the production of essential nutrients used by plants and bacteria.

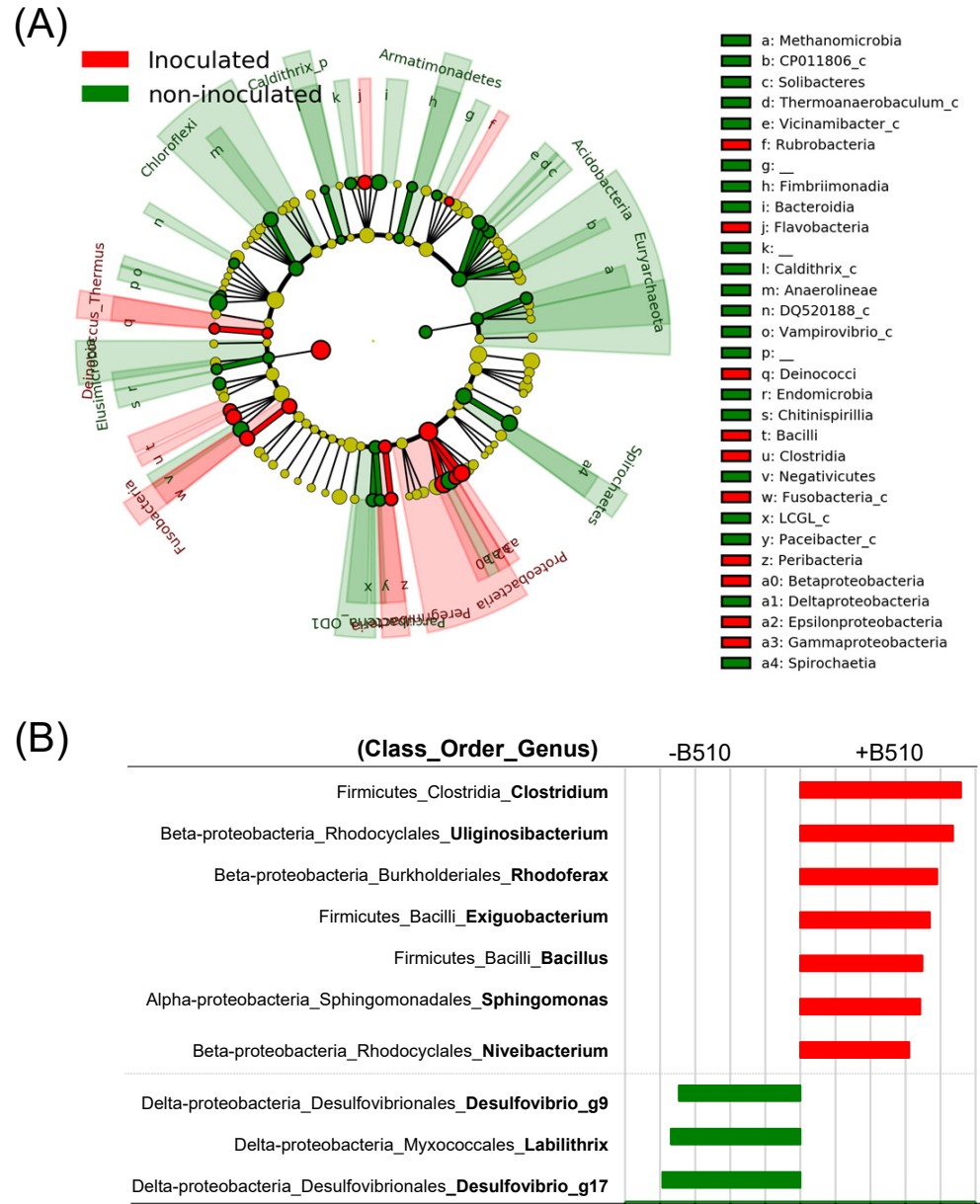

**Figure 3.** Linear discriminant analysis effect size (LEfSe) method identifies the significantly different ($p < 0.05$, Kruskal-Wallis test) bacteria at multiple taxonomic levels by comparing the community composition of non-inoculated and B510-inoculated rice roots in the non-fertilized (−N) condition. (**A**) A bacterial cladogram identifying the taxa with significantly different abundances between treatments are represented by colored Phylum, Class, Order and Family levels. The colored shadows represent trends of the significantly different taxa, B510-inoculated (red) and B510 non-inoculated (green). (**B**) LDA scores showed significant differences resulting from B510-inoculation. The red bars represent the significantly increased taxa, and green bars represent the decreased taxa resulting from B510-inoculation.

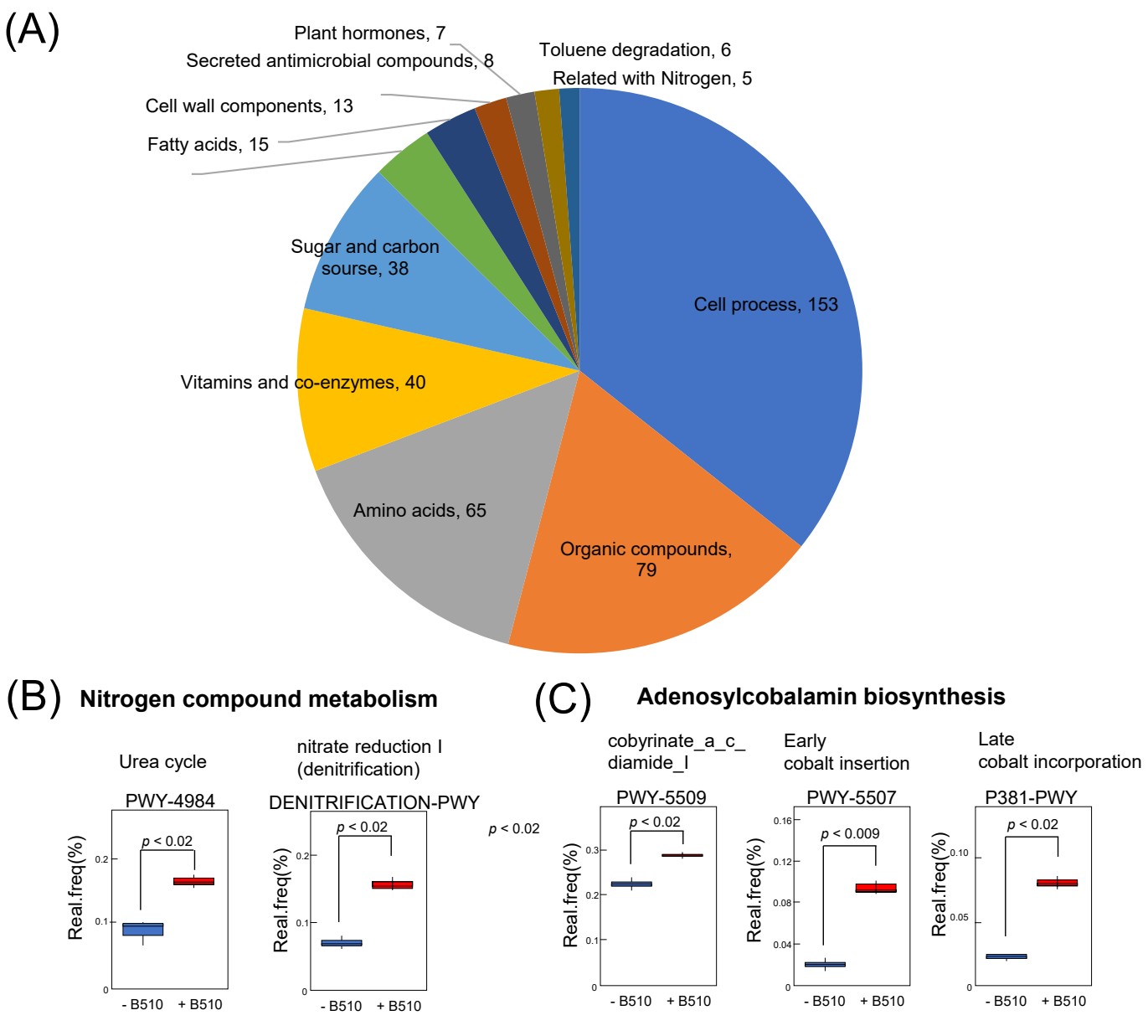

**Figure 4.** Functional composition profiles of bacterial communities. Bacterial communities were analyzed by PICRUSt2. Rice samples with significant differences ($p < 0.05$) between those inoculated with and without B510 in the non-nitrogen fertilized condition. (**A**) Number of pathways selected in each category by PICRUSt2. (**B**) Real frequency (%) of nitrogen compound metabolism, and (**C**) real frequency (%) of cobalamin-related pathways as a function of B510 treatment.

### 3.4. The Effect of B510 Inoculation on Fungal Community Structure

To identify the effect of B510 inoculation on the fungal community structure, we analyzed amplicons using the fungal ITS primer. The $\alpha$-diversity analysis showed that the Shannon and Chao1 index was not significantly different among the treatments (Figure 5A, Table S3). However, there was a significant difference between the vegetative and harvest stages in the Chao1 index (Table S3). Although B510 inoculation affected the bacterial $\alpha$-diversity, the fungal $\alpha$-diversity was not affected by nitrogen application or B510 inoculation. The beta-diversity of PCoA indicated that the samples were divided into two groups, the vegetative stage and the harvest stage, but there was no apparent difference among the treatments (Figure 5B). The heatmap showed that *Sordariomycetes* were abundant in all of the treatments (more than 68% in the vegetative stage and more than 50% in the harvested stage). Although the relative abundance of *Agaricomycetes* was highest in the

non-inoculated −N condition (16.5%) in the vegetative stage, the levels of this fungus in the B510 inoculation and +N treatments were much lower (4.5%, 5% and 3.6%, respectively) (Figure 6). In contrast, the relative abundance of *Ustilaginomycetes* was 0.5% in the non-inoculated −N condition, but that value increased in the B510 inoculation treatment (1.1%), the non-inoculated (+N) treatment (0.9%) and the B510-inoculated (+N) condition (1.7%) without significant differences (Figure 6). To identify the unique fungal taxa that are significantly related to B510 inoculation, the fungal community was depicted using an LDA score >2.5 (Figure S3). The abundance of *Amanita*, *Pseudeurotium*, *Cordyceps*, *Coinella* and *Valsonectria* increased as a result of B510 inoculation.

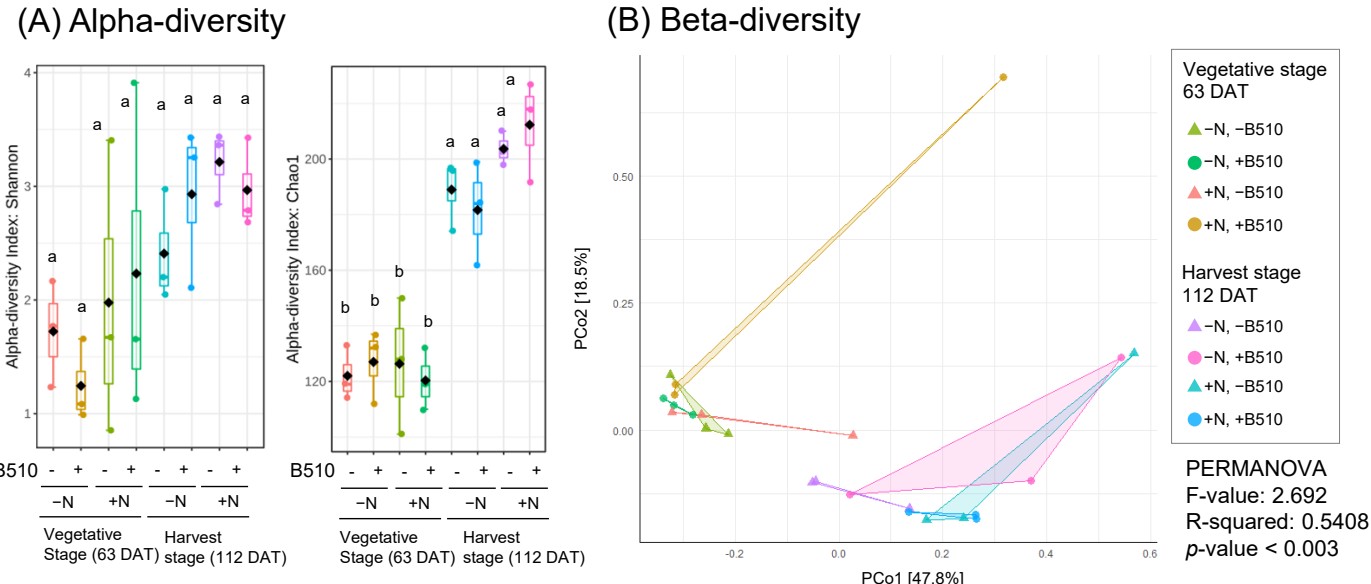

**Figure 5.** Influence of B510 on root-colonized microbiota with different levels of N-fertilization and two stages of plant development. Measurement of the alpha-diversity (**A**) and beta-diversity (**B**) of the rice root-colonized fungi. Rice plants inoculated with B510 (+) and without B510 (−), and fertilized with chemical nitrogen (+N) and non-fertilized (−N) harvested at the vegetative stage (63 DAT) and the harvest stage (112 DAT). (**A**) Shannon index and Chao1 index were used in the analysis of the alpha diversity. All values are means of three replicates ± standard errors. Statistical analysis was performed, different letters indicate significant difference between treatments (Student-Newman-Keuls [SNK] test, $p < 0.05$, n = 3). (**B**) Principal Coordinate Analysis (PCoA) are plotted based on Bray-Curtis distance metrices for taxonomical data ($p < 0.01$). Permutational multivariate analysis of variance (PERMANOVA) was performed.

The heat-maps show the relative abundances of fungal operational taxonomic units (OTUs) at the class level from rice plants inoculated with B510 (+) and without B510 (−) and with supplemental nitrogen fertilizer (+N) and without N-fertilizer (−N). The color of the heat-map indicates the relative abundance from high (red) to low (blue). The asterisks identify significant differences between B510-inoculated (+) and without B510 (−) analyzed by the unpaired two-tailed Student's t-test (* $p < 0.05$, ** $p < 0.01$).

| | Vegetative stage (63 DAT) | | | | Harvest stage (112 DAT) | | | |
|---|---|---|---|---|---|---|---|---|
| Sordariomycetes | 68.1 | 83.1 | 74.9 | 82.7 | 63.2 | 52.2 | 57.6 | 50.9 |
| Agaricomycetes | 16.5 | 4.5 | 5 | 3.6 | 14.6 | 11.1 | 13.7 | 22.7 |
| Dothideomycetes | 1.1 | 2.9 | 2.9 | 3.9 | 7.8 | 19.3 | 12.9 | 11.9 |
| Ascomycota OTU_004 | 0.1 | 0.1 | 0.2 | 0.2 | 5.5 | 8.1 | 7.7 | 7.3 |
| Leotiomycetes | 2.2 | 1.8 | 4.4 | 2 | 0.5 | 1 | 1.1 | 1.1 |
| Ustilaginomycetes | 0.5 | 1.1 | 0.9 | 1.7 | 0.4 | 1.3 | 0.5 | 0.3 |
| Saccharomycetes | 1 | 0.8 | 0.7 | 2.1 | 0.6 | 0.4 | 0.4 | 0.2 |
| Fungi OTU_021 | 0.2 | 0.2 | 1 | 0.1 | 2.2 | 0.4 | 0.1 | 0.2 |
| Fungi OTU_023 | 1.3 | 0.4 | 0.9 | 0.1 | 0.2 | 0.3 | 0.3 | 0.1 |
| Monoblepharidomycetes | 0.4 | 0.3 | 1.1 | 0.5 | 0.2 | 0.4 | 0.3 | 0.2 |
| Ascomycota OTU_030 | 0 | 0 | 0 | 0 | 0.8 | 0.5 | 0.8 | 0.6 |
| Fungi OTU_037 | 0 | 0 | 2.4 | 0 | 0 | 0.2 | 0 | 0 |
| Fungi OTU_038 | 1.8 | 0 | 0 | 0 | 0 | 0 | 0.3 | 0 |
| Ascomycota OTU_039 | 0 | 0 | 0 | 0 | 0.1 | 0.1 | 0.5 | 1.1 |
| Tremellomycetes | 0.1 | 0.1 | 0.3 | 0.2 | 0.2 | 0.3 | 0.4 | 0.2 |
| Fungi OTU_058 | 0 | 0 | 0.9 | 0 | 0.5 | 0.1 | 0.1 | 0.1 |
| Fungi OTU_054 | 0.4 | 0.1 | 0.2 | 0 | 0.3 | 0.6 | 0 | 0 |
| Mortierellomycetes | 0.2 | 0.1 | 0.2 | 0.3 | 0.1 | 0.2 | 0.3 | 0.3 |
| Fungi OTU_057 | 0.8 | 0 | 0 | 0.2 | 0 | 0 | 0.5 | 0 |
| Fungi OTU_065 | 0 | 0.1 | 0.1 | 0.2 | 0.1 | 0.3 | 0.2 | 0.2 |
| B510 | − | + | − | + | − | + | − | + |
| | −N | | +N | | −N | | +N | |

**Figure 6.** Heat Map of the top 20 fungal species associated with rice roots at the vegetative stage (63 DAT) and harvest stage (112 DAT).

## 4. Discussion

This study demonstrated a marked shift in the bacterial community resulting from B510 inoculation in only the −N condition at the vegetative stage (Figure 3A). However, there was a slight community shift as a result of B510 inoculation and nitrogen application at the harvest stage. Additionally, the fungal community structure was not affected by B510 inoculation (Figure 3B). Several studies had already shown that B510 has an $N_2$-fixing ability; rice plants inoculated with this strain showed enhanced growth resulting from B510 inoculation. The plant-growth-promoting effects of B510 were previously observed in low nitrogen conditions [18,29]. Our report is the first to show a significant bacterial community shift resulting from B510 inoculation under non-applied nitrogen conditions when supplemental nitrogen was not present.

### 4.1. Impact of Azospirillum Inoculation on Fields

*Azospirillum* is a well-known biofertilizer; however, there are only a few published reports documenting the effect of *Azospirillum* on plant root communities. Previous studies indicated that the *Azospirillum* inoculant affected the bacterial communities of the maize and paddy rice rhizospheres. Prior to our work, there was only one report in which the same strain, B510, and the same rice cultivar used in our study were tested [5]. B510 inoculation effected a minor change in rice-associated bacteria in conditions where nitrogen was applied. Alpha-proteobacteria (>30%), *Rhizobium* (9.5–20.7%) and *Methylobacterium* (15.2–22%) were dominant in the base of the rice. In our study, the heat map (Figure 2) showed differences in the relative abundances of beta-proteobacteria (11.8–28%), delta-proteobacteria (14.9–26.8%) and alpha-proteobacteria (5–22.3%). Clearly, the location and field conditions affected the rice rhizosphere's community structure. Some taxa had a similar tendency to be affected by B510 inoculation, leading to an increase in mainly Firmicutes (*Clostridia* and *Bacilli*) and beta-proteobacteria (Burkholderiales) (Figure 3B). The inoculation of maize with *Azospirillum brasilense* did not affect the specific rhizobacterial community [30]. In contrast, the inoculation with *Azospirillum lipoferum* CRT1 significantly impacted the total bacterial community in maize fields [31]. Changes in the microbial community resulting from *Azospirillum* inoculation have not yet been reported for other plant species.

### 4.2. Inoculation with Azospirillum sp. B510 Increased Nitrogen-Fixing Bacteria

The LeFSE analysis showed an increase in the typical diazotrophic bacteria *Clostridium*, *Bacillus* and *Sphingomonas* as a result of B510 inoculation (Figure 3B). *Clostridium* is an anaerobic gram-positive bacterium belonging to the phylum Firmicutes. *Clostridium* was known as PGPR of rice, clover and some other crops [32,33]. Lu et al. (2006) [34] suggested that *Clostridia* inhabiting rice roots and the rhizosphere are probably responsible for the anaerobic decomposition of decaying root residues. The *Bacillus* species are known to secrete antimicrobial active substances; the increased ratio of these substances may have contributed to the enhanced disease resistance of rice inoculated with B510 [22].

*Aeromonas* is a genus of gram-negative, facultative anaerobic and rod-shaped bacteria that morphologically resemble members of the family Enterobacteriaceae and are important human pathogens. *Aeromonas* also has been reported as a PGPR and/or biocontrol agent from the rhizosphere of several crops, including rice, wheat and beans. *Aeromonas* has the NifH gene, giving the bacterium the potential to fix nitrogen [35]. Nitrogen-fixing *Sphingomonas* was isolated from rice in Brazil and Japan [36,37]. *Uliginosibacterium* is gram-negative, strictly aerobic and rod-shaped beta-proteobacterium. The analysis of 16S rRNA showed that *Uliginosibacterium* is closely related to *Azoarcus* [38]; however, the NifH gene of *U. gangwonense* is most similar to that of *Formivibrio citricus* and belongs to NifH cluster I, the same group to which *Bradyrhizobium* and *Paraburkholderia* belong [39]. Hydrogen-oxidizing bacteria, *Hydrogenophaga* sp., were ascertained to benefit soybean growth and salinity tolerance [40]. *Exiguibacterium* is also known as a PGPR [41]; however, the increment of nitrogen-fixing bacterial abundance caused by B510 inoculation and the direct relationship between B510 and the selected bacteria is not clear. B510 inoculation altered the secondary metabolite profiles of rice, including flavonoids and phenolic acids [16]. The change in the metabolite profiles secreted from rice roots was proposed to have caused the marked shift in the microbial community. Moreover, B510 inoculation also increased non-PGPR members: *Rhodoferax* and *Niveibacterium*. *Rhodoferax* was found to be the most abundant genus in the maize rhizosphere [42]. Until now, neither of these bacteria showed any PGPR activity, so additional studies are needed to show that these genera can improve rhizosphere conditions and enhance plant growth.

### 4.3. The Function of Cobalamin in the Rhizosphere Community

This study predicted that cobalamin might be a functional compound in soil produced by bacteria induced by rice plants colonized with B510. In addition to B510, more than 600 bacterial species, including *Pseudomonas*, *Bacillus*, *Burkholderia*, *Paraburkhorderia*, *Roseomonas*, *Rhizobia* and *Fuscibacter,* have a CobW gene in their genomes responsible for cobalamin biosynthesis [43–45]. Many bacteria in the plant rhizosphere produce cobalamin, and cobalamin stimulates non-cobalamin-synthesis bacterial growth [46]. Additionally, B510 has 20 genes related to cobalamin biosynthesis (Table S5 [20]). Moreover, plants are incapable of producing cobalamin (Smith et al., 2007). The function of cobalamin in plants is not clear; however, cobalamin is an essential factor for the symbiotic interaction between *Shinorhizobium meliloti* and *Medicago sativa* (alfalfa) [47]. Moreover, *Lepidium sativum* (garden cress) can absorb and transport cobalamin, which localizes to vacuoles in the cotyledons [48]. Interestingly, *Bacteroides* localized inside human gastrointestinal tracts secrete polysaccharides in cooperation with other bacteria [49]. In the case of PGPRs, *Bacillus velezensis* produces branched-chain amino acids (BCAAs) for other PGPRs [50]. It seems like a similar ecosystem is working within the rice rhizosphere. These studies indicated that cobalamin production by bacteria functions as a cooperation compound within the rhizosphere microbiota. These results support the hypothesis that *Azospirillum* also contributes to other microbes by metabolic cross-feeding.

In conclusion, our findings demonstrated that the inoculation with B510 recruits indigenous beneficial $N_2$-fixing bacteria in the rice rhizosphere in paddy fields. Even if supplemental nitrogen was not applied to the paddies, the rice plants grew better, as if they were fertilized with chemical nitrogen, and a marked shift in the microbial community

structure occurred. In our study, we did not identify how B510 recruits other N$_2$-fixing bacteria. However, when rice plants are grown in an aseptic condition, the amount of malic acid, a typical organic acid secreted by roots to recruit beneficial soil bacteria such as *Bacillus* [51], increased more than 2.5 times after B510 inoculation [29]. In addition, *Azospirillum* inoculation influenced the metabolite production by rice plants, including such compounds as flavonoids and hydroxycinnamic derivatives [16]. Additionally, the modification of rice metabolites by B510 inoculation caused a marked change in the microbial community structure in the rice rhizosphere. Further research will reveal additional facets as to why *Azospirillum* is useful as a biofertilizer.

**Supplementary Materials:** The following supporting information can be downloaded at: https://www.mdpi.com/article/10.3390/agronomy12061367/s1, Figure S1: Rarefaction curve of bacteria (A) and Fungi (B), Figure S2: Ab-solute differences in relative abundance pathways between the non-treated control and B510-inoc-ulated rhizosphere samples, Figure S3: LEfSe analysis of the fungal genus, Table S1: Soil chemical properties of the rice paddy field, Table S2: Read number of the amplicon analysis in this study, Table S3: Alpha-diversity indices of the bacterial and fungal communities of rice root inoculated with *Azospirillum* sp. B510, Table S4: Statistical table of pathway from PICRUSt2, Table S5: Adenosylcobalamin-related genes in *Azospirillum* sp. B510.

**Author Contributions:** Conceptualization, M.Y. and S.O.; methodology, M.Y.; validation, M.Y.; formal analysis, M.Y.; investigation, M.Y., C.T. and E.S.-A.; resources, T.I., S.S. and S.O.; writing—original draft preparation, M.Y.; writing—review and editing, S.O. and K.M.G.D.; visualization, M.Y.; supervision, S.O.; project administration, S.O.; funding acquisition, M.Y. and S.O. All authors have read and agreed to the published version of the manuscript.

**Funding:** This research was supported by the Grant-in Aid for JSPS Fellows (20J40199 to M.Y.) and the JSPS Bilateral Program (120199926 to S.O.).

**Institutional Review Board Statement:** Not applicable.

**Informed Consent Statement:** Not applicable.

**Data Availability Statement:** Not applicable.

**Acknowledgments:** We thank Sakiko Ueda (Woman Support Center, Tokyo University of Agriculture and Technology) for the technical support. We highly appreciate Keisuke Katsura (Tokyo University of Agriculture and Technology), who helps us manage the field experiment. We would also like to thank Akiko Yoshida and Appiah Kwame and the colleagues who helped and supported gathering experimental data and technical support.

**Conflicts of Interest:** The authors declare no conflict of interest.

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
