# Peer review of "Impact of Azospirillum sp. B510 on the Rhizosphere Microbiome of Rice under Field Conditions"

_agronomy, doi:10.3390/agronomy12061367_

Round 1

Reviewer 1 Report

Comments on the ms entitled “Impact of Azospirillum sp. B510 on the rhizosphere microbiome of rice under field conditions”

Overall
The topic of the manuscript is of high interest of international readership. Exploring the potential ability of microorganisms to colonize the soil, improve plant growth and development, and increase agricultural yield while respecting the environment is highly important issue. The reviewed work is innovative, its Authors are the first to show significant bacterial community shift resulting from Azospirillum sp. B510-inoculation under non-applied nitrogen conditions when supplemental nitrogen was not present. Like the results for changes in the microbial community resulting from Azospirillum-inoculation have not yet been reported for other plant species. I find the results on functional characteristics of microorganisms described from rice rhizosphere after inoculation very interesting and promising for the future.

Undertaking research on this topic is an important aspect of this work. The results obtained are interesting and should be continued.

I have some minor comments regarding the layout of the paper and the presentation of the research results, which I have included in the text of the article.

Author Response

Manuscript ID: Agronomy-1731167

Response to Reviewer 1

We would like to express our appreciation to the reviewer for the discerning comments and suggestions, which have helped us to improve this paper.

  1. Article line 15: numerous articles have been published concerning the effects of biofertilizers on soil microbial communities.

Response: I agreed reviewer’s comment, and I changed this sentence as below:

Before: Little is known about the influence of biofertilizers on the composition of microbial communities associated with crop plants.

After: Increasing attention about the influence of biofertilizers on the composition of microbial communities associated with crop plants.

  1. Article lines 61 to 63: the scientific questions raised in this article are very good, but they have little to do with the content of the article, and the research done in the article cannot fully explain the questions raised.

Response: In this sentence, we just described about the situation of this research before our study. I think we clarified a part of the relationships the importance of the N2-fixing bacteria induced by the B510-inoculation. So, I this we don’t need remove this sentence.

  1. Article lines 192 to 194: the first half of this sentence refers to the processing of adding N, and the second half of the sentence refers to the processing of not adding N. Is it appropriate to use "in which" in the middle to keep things coherent?

Response: I agreed the reviewer’s comment, and I changed this sentence as below:

Before: The most abundant taxa was beta-proteobacteria in the +N condition at the vegetative stage (-N+B510) in which the relative abundance of beta-proteobacteria increased from 11.8% to 27% after B510 inoculation in the absence of supplemental nitrogen during the vegetative stage.

After: The most abundant taxa was beta-proteobacteria in the +N condition at the vegetative stage. The relative abundance of beta-proteobacteria increased from 11.8% to 27% after B510 inoculation in the absence of supplemental nitrogen (-N) condition during the vegetative stage.

  1. Article lines 210 and 232: the PCoA graph can only see that there is a difference between the two treatments, and cannot see whether the difference is significant. Further testing is needed. The author did not make.

Response: I appreciate this reviewer pointing out our mistakes. I added the statistical analysis of the PCoA in Fig. 1B. 

5.Article lines 214: no images in the article are represented as Fig. 4AB.

Response: I removed Fig. 4A, and I added new figure as supplement (Fig. S2). And I also change this sentence as below: A total of 424 pathways were selected by PICRUSt2 (Fig. S2, Table S4)

  1. Article lines 255: as mentioned later in the article, "Previous studies indicated that Azospirilluminoculant affected the bacterial communities of the maize and paddy rice rhizospheres.", B510 is also Azospirillum sp., is this the first time?

Response: This is the first paper to show the distinct difference affect the microbiota of rice rhizospheres. Azospirillum is very famous PGPR and some of them used for as biofertilizers.

  1. Article lines 286 to 299: There are many bacteria mentioned in this passage, which are not covered in the above results. Whether it is meaningful to discuss here and whether the supporting role of the research results of the article is established?

Response: In this sentence, we mentioned about the bacteria selected by LeFSE analysis. Some of these bacteria were clarified the function to the host plant, and we think these functions contribute to the beneficial effect as PGPR.

  1. Table 1: Some numerical error values are too large, please check whether the data is correct, and please test whether the data marked in yellow are significantly different.

Response: I checked this table again, and I confirmed this statistical analysis was correct.  

  1. Figures 1, 2, 5, and 6: it is recommended to add the legend.

Response: I added the legends in Fig. 1B and Fig. 5B.

  1. Figures 1 and 5: The colors between Figures A and B are the same, but they represent different treatments. Do they need to be consistent between a group of pictures? And the results of PERMANOVA are not seen in Figures 1B and 5B.

Response: I also wanted to changing the figure colors. But, R (package Ampvis2) decided sample color automatically. I could not change these figure color. I added the results of PERMANOVA in both Fig. 1B and 5B.

  1. Figure 3: please use the original image, and the legend font is too small. In Fig. 3A, only three classification levels can be seen, and the figure annotation do not correspond.

Response: I changed Fig. 3A to the original image.

  1. Figure 4: please use the original image, and the legend font is too small, and please indicate significance in Fig. 4C and 4D.

Response: I changed this Fig. 4. Fig. 4A moved to Fig. S2 to see over all. Fig. 4C and 4D (changed to Fig. 4B and 4C) have already cut off by p < 0.05 by PICRUSt2. I added p value in Fig. 4B and 4C.

  1. Figure 5B: poor reproducibility between treatments.

Response: I want to know the reason why the fungal microbiota showed variety. Even though we used the same plot, the fungal community structure was not stable rather than bacteria. I think the one possibility for this reason, there is the difference water level among them. To dissolve this problem, we should make better condition of paddy field before transplantation. In the future, we will consider that problem. Thank you for giving us this suggestion.

Reviewer 2 Report

1. Article line 15: numerous articles have been published concerning the effects of biofertilizers on soil microbial communities.

2. Article lines 61 to 63: the scientific questions raised in this article are very good, but they have little to do with the content of the article, and the research done in the article cannot fully explain the questions raised.

3. Article lines 192 to 194: the first half of this sentence refers to the processing of adding N, and the second half of the sentence refers to the processing of not adding N. Is it appropriate to use "in which" in the middle to keep things coherent?

4. Article lines 210 and 232: the PCoA graph can only see that there is a difference between the two treatments, and cannot see whether the difference is significant. Further testing is needed. The author did not make.

5.Article lines 214: no images in the article are represented as Fig. 4AB.

6. Article lines 255: as mentioned later in the article, "Previous studies indicated that Azospirillum inoculant affected the bacterial communities of the maize and paddy rice rhizospheres.", B510 is also Azospirillum sp., is this the first time?

7. Article lines 286 to 299: There are many bacteria mentioned in this passage, which are not covered in the above results. Whether it is meaningful to discuss here and whether the supporting role of the research results of the article is established?

8. Table 1: Some numerical error values are too large, please check whether the data is correct, and please test whether the data marked in yellow are significantly different.

9. Figures 1, 2, 5, and 6: it is recommended to add the legend.

10. Figures 1 and 5: The colors between Figures A and B are the same, but they represent different treatments. Do they need to be consistent between a group of pictures? And the results of PERMANOVA are not seen in Figures 1B and 5B.

11. Figure 3: please use the original image, and the legend font is too small. In Fig. 3A, only three classification levels can be seen, and the figure annotation do not correspond.

12. Figure 4: please use the original image, and the legend font is too small, and please indicate significance in Fig. 4C and 4D.

13. Figure 5B: poor reproducibility between treatments.

Author Response

Manuscript ID: Agronomy-1731167

Response to Reviewer 

We would like to express our appreciation to the reviewer for the discerning comments and suggestions, which have helped us to improve this paper.

  1. Article line 15: numerous articles have been published concerning the effects of biofertilizers on soil microbial communities.

Response: I agreed reviewer’s comment, and I changed this sentence as below:

Before: Little is known about the influence of biofertilizers on the composition of microbial communities associated with crop plants.

After: Increasing attention about the influence of biofertilizers on the composition of microbial communities associated with crop plants.

  1. Article lines 61 to 63: the scientific questions raised in this article are very good, but they have little to do with the content of the article, and the research done in the article cannot fully explain the questions raised.

Response: In this sentence, we just described about the situation of this research before our study. I think we clarified a part of the relationships the importance of the N2-fixing bacteria induced by the B510-inoculation. So, I this we don’t need remove this sentence.

  1. Article lines 192 to 194: the first half of this sentence refers to the processing of adding N, and the second half of the sentence refers to the processing of not adding N. Is it appropriate to use "in which" in the middle to keep things coherent?

Response: I agreed the reviewer’s comment, and I changed this sentence as below:

Before: The most abundant taxa was beta-proteobacteria in the +N condition at the vegetative stage (-N+B510) in which the relative abundance of beta-proteobacteria increased from 11.8% to 27% after B510 inoculation in the absence of supplemental nitrogen during the vegetative stage.

After: The most abundant taxa was beta-proteobacteria in the +N condition at the vegetative stage. The relative abundance of beta-proteobacteria increased from 11.8% to 27% after B510 inoculation in the absence of supplemental nitrogen (-N) condition during the vegetative stage.

  1. Article lines 210 and 232: the PCoA graph can only see that there is a difference between the two treatments, and cannot see whether the difference is significant. Further testing is needed. The author did not make.

Response: I appreciate this reviewer pointing out our mistakes. I added the statistical analysis of the PCoA in Fig. 1B. 

5.Article lines 214: no images in the article are represented as Fig. 4AB.

Response: I removed Fig. 4A, and I added new figure as supplement (Fig. S2). And I also change this sentence as below: A total of 424 pathways were selected by PICRUSt2 (Fig. S2, Table S4)

  1. Article lines 255: as mentioned later in the article, "Previous studies indicated that Azospirilluminoculant affected the bacterial communities of the maize and paddy rice rhizospheres.", B510 is also Azospirillum sp., is this the first time?

Response: This is the first paper to show the distinct difference affect the microbiota of rice rhizospheres. Azospirillum is very famous PGPR and some of them used for as biofertilizers.

  1. Article lines 286 to 299: There are many bacteria mentioned in this passage, which are not covered in the above results. Whether it is meaningful to discuss here and whether the supporting role of the research results of the article is established?

Response: In this sentence, we mentioned about the bacteria selected by LeFSE analysis. Some of these bacteria were clarified the function to the host plant, and we think these functions contribute to the beneficial effect as PGPR.

  1. Table 1: Some numerical error values are too large, please check whether the data is correct, and please test whether the data marked in yellow are significantly different.

Response: I checked this table again, and I confirmed this statistical analysis was correct.  

  1. Figures 1, 2, 5, and 6: it is recommended to add the legend.

Response: I added the legends in Fig. 1B and Fig. 5B.

  1. Figures 1 and 5: The colors between Figures A and B are the same, but they represent different treatments. Do they need to be consistent between a group of pictures? And the results of PERMANOVA are not seen in Figures 1B and 5B.

Response: I also wanted to changing the figure colors. But, R (package Ampvis2) decided sample color automatically. I could not change these figure color. I added the results of PERMANOVA in both Fig. 1B and 5B.

  1. Figure 3: please use the original image, and the legend font is too small. In Fig. 3A, only three classification levels can be seen, and the figure annotation do not correspond.

Response: I changed Fig. 3A to the original image.

  1. Figure 4: please use the original image, and the legend font is too small, and please indicate significance in Fig. 4C and 4D.

Response: I changed this Fig. 4. Fig. 4A moved to Fig. S2 to see over all. Fig. 4C and 4D (changed to Fig. 4B and 4C) have already cut off by p < 0.05 by PICRUSt2. I added p value in Fig. 4B and 4C.

  1. Figure 5B: poor reproducibility between treatments.

Response: I want to know the reason why the fungal microbiota showed variety. Even though we used the same plot, the fungal community structure was not stable rather than bacteria. I think the one possibility for this reason, there is the difference water level among them. To dissolve this problem, we should make better condition of paddy field before transplantation. In the future, we will consider that problem. Thank you for giving us this suggestion.
